# BOOTSTRAPPING EXPECTILES IN ROBUST REINFORCEMENT LEARNING

## ABSTRACT

Many classic Reinforcement Learning (RL) algorithms rely on a Bellman operator, which involves an expectation over the next states, leading to the concept of bootstrapping. To introduce a form of pessimism, we propose to replace this expectation with an expectile. In practice, this can be very simply done by replacing the $L_2$ loss with a more general expectile loss for the critic. Introducing pessimism in RL is desirable for various reasons, such as tackling the overestimation problem (for which classic solutions are double Q-learning or the twin-critic approach of TD3) or robust RL (where transitions are adversarial). We study empirically these two cases. For the overestimation problem, we show that the proposed approach, `ExpectRL`, provides similar results to a classic twin-critic. On robust RL benchmarks, involving changes of the environment, we show that our approach is more robust than classic RL algorithms. We also introduce a variation of `ExpectRL` combined with domain randomization which is competitive with state-of-the-art robust RL agents. Eventually, we also extend `ExpectRL` with a mechanism for choosing automatically the expectile value, that is the degree of pessimism.

## 1 INTRODUCTION

Pessimism is a desirable concept in many Reinforcement Learning (RL) algorithms to stabilize the learning and get an accurate estimation of the value function. This idea is developed in Double Q-learning (Hasselt, 2010), an RL technique designed to address the issue of overestimation bias in value estimation, a common challenge in Q-learning and related algorithms. Overestimation bias occurs when the estimated values of actions are higher than their true values, potentially leading to a suboptimal policy. By maintaining two sets of Q-values and decoupling action selection from value estimation, Double Q-learning provides a more accurate and less optimistic estimate of the true values of actions. In general, Double Q-learning enhances the stability of the learning process and these principles can be extended to deep RL known as Double Deep Q-Networks (DDQN), a successful approach in various applications (Van Hasselt et al., 2016). Pessimism also appears in the twin critic approach, the equivalent of Double Q-learning for continuous action spaces, which requires training two critics to select the most pessimistic one. Many state-of-the-art RL algorithms are based on this method, such as TD3 (Fujimoto et al., 2018) that uses this method to improve on DDPG (Lillicrap et al., 2015) and SAC (Haarnoja et al., 2018) that uses this trick to stabilize the learning of $Q$-functions and policies.

The idea of pessimism is also central in Robust RL (Moos et al., 2022), where the agent tries to find the best policy under the worst transition kernel in a certain uncertainty space. It has been introduced first theoretically in the context of Robust MDPs (Iyengar, 2005; Nilim & El Ghaoui, 2005) (RMDPs) where the transition probability varies in an uncertainty (or ambiguity) set. Hence, the solution of robust MDPs is less sensitive to model estimation errors with a properly chosen uncertainty set, as RMDPs are formulated as a max-min problem, where the objective is to find the policy that maximizes the value function for the worst possible model that lies within an uncertainty set around a nominal model. Fortunately, many structural properties of MDPs are preserved in RMDPs (Iyengar, 2005), and methods such as robust value iteration, robust modified policy iteration, or partial robust policy iteration (Ho et al., 2021) can be used to solve them. It is also known that the uncertainty in the reward can be easily tackled while handling uncertainty in the transition kernel is much more difficult (Kumar et al., 2022; Derman et al., 2021). Finally, the sample complexity of RMDPs has been

studied theoretically (Yang et al., 2021; Panaganti et al., 2022; Clavier et al., 2023; Shi et al., 2023; Clavier et al., 2025). However, these works usually assume having access to a generative model.

Robust RL (Moos et al., 2022) tries to bridge a gap with real-life problems, classifying its algorithms into two distinct groups. The first group engages solely with the nominal kernel or the center of the uncertainty set. To enhance robustness, these algorithms often adopt an equivalent risk-averse formulation to instill pessimism. For instance, Clavier et al. (2022) employ mean-standard deviation optimization through Distributional Learning to bolster robustness. Another strategy involves introducing perturbations on actions during the learning process, as demonstrated by Tessler et al. (2019), aiming to fortify robustness during testing. Another method, known as adversarial kernel robust RL (Wang et al., 2023), exclusively samples from the nominal kernel and employs resampling techniques to simulate the adversarial kernel. While this approach introduces a novel paradigm, it also leads to challenges associated with poor sample complexity due to resampling and requiring access to a generative model. Despite this drawback, the adversarial kernel robust RL paradigm offers an intriguing avenue for exploration and development in the realm of robust RL. Finally, policy gradient (Kumar et al., 2023; Li et al., 2023) in the case of Robust MDPs is also an alternative. A practical algorithm using robust policy gradient with Wasserstein metric is proposed by Abdullah et al. (2019), but this approach requires having access to model parameters which are usually not available in a model-free setting. The second category of algorithms engages with samples within the uncertainty set, leveraging available information to enhance the robustness and generalization of policies to diverse environments. Algorithms within this category, such as IWOCS (Zouitine et al., 2023), M2TD3 (Tanabe et al., 2022), M3DDPG (Li et al., 2019), and RARL (Pinto et al., 2017) actively interact with various close environments to fortify robustness in the context of RL.

In all these settings, the idea of pessimism is central. We propose here a new simple form of pessimism based on expectile estimates that can be plugged into any RL algorithm. For a given algorithm, the only modification relies on the critic loss in an actor-critic framework or in the $Q$-learning loss for $Q$-function based algorithms. Given a target $y(r, s') = r + \gamma Q_{\phi_{\mathrm{targ}}}(s', \pi(s'))$ with reward $r$, policy $\pi$, we propose to minimize $L(\phi, \mathcal{D}) = \underset{(s,a,r,s') \sim \mathcal{D}}{\mathbb{E}} \left[ L_2^\alpha \left( Q_\phi(s, a) - y(r, s') \right) \right]$,

where $L_2^\tau$ is the expectile loss defined in Section 3.3. For $\alpha = 1/2$, the expectile coincides with the classical mean, and we recover the classical $L_2$ loss of most RL algorithms. We denote this modification as `ExpectRL`. In many RL algorithms, we are bootstrapping the expectation of the $Q$-function over the next state, by definition of the classical Bellman equation. Our method `ExpectRL` is equivalent to bootstrapping the expectile and not the expectation of the Q value. Bootstraping expectiles still leads to an algorithm with the contraction mapping property for the associated Expectile Bellman Operator, but adds pessimism by giving more weight to the pessimistic next state compared to a classical expectation (see Section 3.3).

The `ExpectRL` modification is relevant in the context of the twin critic approach as when employing this method, the challenge arises in effectively regulating the level of pessimism through the application of the twin critic method, which remains heuristic for continuous action spaces, although it has been studied in the discrete case by Hasselt (2010). Furthermore, the acquisition of imprecise $Q$ functions has the potential to yield detrimental outcomes in practical applications, introducing the risk of catastrophic consequences. Using the `ExpectRL` method, the degree of pessimism in learning the value or $Q$ function is controlled through the parameter $\alpha$, and our first question is:

*Can we replace the learning of two critics in the twin critic method, using only a simple expectile bootstrapping?*

In the Robust RL setting, `ExpectRL` can also be beneficial as by nature expectiles are a coherent, convex risk measure, that can be written as a minimum of an expectation over probability measure on a close convex set (Delbaen, 2002). So implicitly bootstrapping an expectile instead of an average leads to a robust RL algorithm. Compared to many Robust RL algorithms, our method is simple in the sense that the $\alpha$-expectile is more interpretable and easy to choose than a penalization or trade-off parameter in mean-standard deviation optimization (Clavier et al., 2022). `ExpectRL` has the advantage of being computationally simple compared to other methods, as it uses all samples, compared to the work of Wang et al. (2023), that needs resampling to induce robustness. Finally, our method is simple and can be adapted to practical algorithms, compared to robust policy gradient methods such as Kumar et al. (2023); Li et al. (2023). Moreover, while these algorithms can be considered more mathematically grounded and less heuristic, the second group with IWOCS, M2DTD2, RARL

(Zouitine et al., 2023; Tanabe et al., 2022; Li et al., 2019; Pinto et al., 2017) tends to rely on heuristic approaches that exhibit practical efficacy on real-world benchmarks. This dichotomy prompts the question:

*Can we leverage* `ExpectRL` *method as a surrogate for Robust RL and formulate robust RL algorithms that are both mathematically founded and requiring minimal parameter tuning?*

By extending expectile bootstrapping (`ExpectRL`) with sampling from the entire uncertainty set using domain randomization (DR), our approach bolsters robustness, positioning itself competitively against the best-performing algorithms. Notably, our algorithm incurs low computational costs relatively to other algorithms and requires minimal or no hyperparameter tuning. Our contributions are the following.

Our **first contribution**, is to introduce `ExpectRL`, and demonstrate the efficacy of that method as a viable alternative to the twin critic trick with $L_2$ loss across diverse environments. The **second contribution** of our work lies in establishing that expectile bootstrapping or `ExpectRL` facilitates the development of straightforward Deep Robust RL approaches. These approaches exhibit enhanced robustness compared to classical RL algorithms. The effectiveness of our approach combining `ExpectRL` with DR is demonstrated on various benchmarks and results in an algorithm that closely approaches the state of the art in robust RL, offering advantages such as lower computational costs and minimal hyperparameters to fine-tune. Our **third contribution** introduces an algorithm, `AutoExpectRL` that leverages an automatic mechanism for selecting the expectile or determining the degree of pessimism. Leveraging bandit algorithms, this approach provides an automated and adaptive way to fine-tune the expectile parameter, contributing to the overall efficiency and effectiveness of the algorithm.

## 2 RELATED WORK

**TD3 and twin critics.** To tackle the problem of over-estimation of the value function, TD3 algorithm (Fujimoto et al., 2018) algorithm uses two critics. Defining the target $y_{min}$ as $y_{min}(r, s') = r + \gamma \min_{i=1,2} Q_{\phi_{i,\text{targ}}}(s', \pi(s'))$, both critics are learned by regressing to this target, such that, for $i \in \{1, 2\}$, $L(\phi_i, \mathcal{D}) = \mathbb{E}_{(s,a,r,s',d)\sim\mathcal{D}} (Q_{\phi_i}(s, a) - y_{min}(r, s'))^2$ Our approach is different as we do not consider the classic $L_2$ loss and only use one critic. We will compare `ExpectRL` to the classic TD3 algorithm both with twin critics and one critic to understand the influence of our method.

**Expectiles in Distributional RL.** Expectiles have found application within the domain of Distributional RL (RL), as evidenced by studies such as (Rowland et al., 2019; Dabney et al., 2018; Jullien et al., 2023). It is crucial to note a distinction in our approach, where we specifically focus on learning a single expectile to substitute the conventional $L_2$ norm. This diverges from the methodology adopted in these referenced papers, where the entire distribution is learned using different expectiles. Moreover, they do not consider expectile statistics on the same random variable as they consider expectiles of the full return.

**Expectile in Offline RL and the IQL algorithm .** Implicit Q-learning (IQL) (Kostrikov et al., 2021) in the context of offline RL endeavors to enhance policies without the necessity of evaluating actions that have not been encountered. Like our method, IQL employs a distinctive approach by treating the state value function as a random variable associated with the action, but achieves an estimation of the optimal action values for a state by utilizing a state conditional upper expectile. In `ExpectRL`, we employ lower expectiles to instill pessimism on the next state and approximate a minimum function, contrasting with the conventional use of upper expectiles for approximating the maximum in the Bellman optimality equation.

**Risk-Averse RL.** Risk-averse RL, as explored in studies like Pan et al. (2019), diverges from the traditional risk-neutral RL paradigm. Its objective is to optimize a risk measure associated with the return random variable, rather than focusing solely on its expectation. Within this framework, Mean-Variance Policy Iteration has been considered for optimization, as evidenced by Zhang et al. (2021), and Conditional Value at Risk (CVaR), as studied by Greenberg et al. (2022). The link between Robust and Risk averse MDPS has been highlighted by Chow et al. (2015) and Zhang et al. (2023) who provide a mathematical foundation for risk-averse RL methodologies, emphasizing the

significance of coherent risk measures in achieving robust and reliable policies. Our method lies in risk-averse RL as expectiles are a coherent risk measure (Zhang et al., 2023), but to the best of our knowledge, the expectile statistic has never been considered before for tackling robust RL problems.

**Regularisation and robustness in RL.** Regularization plays a pivotal role in the context of Markov Decision Processes (MDPs), as underscored by Derman et al. (2021) or Eysenbach & Levine (2021), who have elucidated the pronounced connection between robust MDPs and their regularized counterparts. Specifically, they have illustrated that a regularised policy during interaction with a given MDP exhibits robustness within an uncertainty set surrounding the MDP in question. In this work, we focus on the idea that generalization, regularization, and robustness are strongly linked in RL or MDPs as shown by Husain et al. (2021); Derman & Mannor (2020); Derman et al. (2021); Ying et al. (2021); Brekelmans et al. (2022). The main drawback of this method is that it requires tuning the introduced penalization to improve robustness, which is not easy in practice as it is very task-dependent. The magnitude of the penalization is not always interpretable compared to $\alpha$, the value of the expectile.

## 3 BACKGROUND

### 3.1 MARKOV DECISION PROCESSES

We first define Robust Markov Decision Processes (MDPs) as $\mathcal{M}_\Omega = \{\mathcal{M}_\omega\}_{\omega \in \Omega}$, with $\mathcal{M}_\omega = \langle S, A, P_\omega, P_\omega^0, r_\omega, \gamma \rangle$ the MDP with specific uncertainty parameter $\omega \in \Omega$. The chosen state space $S$ and action space $A$ are subsets of real-valued vector spaces in our setting. The transition probability density $P_\omega : S \times A \times S \to \mathbb{R}$, the initial state probability density $P_\omega^0 : S \to \mathbb{R}$, and the immediate reward $r_\omega : S \times A \to \mathbb{R}$ depend on $\omega$. Moreover, we define $P_{sa,w}$ the vector of $P_\omega(s, a, .)$. The discount factor is denoted by $\gamma \in (0, 1)$. Let $\pi_\theta : S \to A$ be a policy parameterized by $\theta \in \Theta$ and $\pi^*$ the optimal policy. Given an uncertainty parameter $\omega \in \Omega$, the initial state follows $s_0 \sim P_\omega^0$. At each time step $t \geqslant 0$, the agent observes state $s_t$, selects action $a_t = \pi_\theta(s_t)$, interacts with the environment, and observes the next state $s_{t+1} \sim P_\omega(\cdot \mid s_t, a_t)$, and the immediate reward $r_t = r_\omega(s_t, a_t)$. The discounted return of the trajectory starting from time step $t$ is $R_t = \sum_{k \geqslant 0} \gamma^k r_{t+k}$. The action value function $q^{\pi_\theta}(s, a, \omega)$ and optimal action value $q^*(s, a, \omega)$ under $\omega$ is the expectation of $R_t$ starting with $s_t = s$ and $a_t = a$ under $\omega$; that is,

$$q^{\pi_\theta}(s, a, \omega) = \mathbb{E}_{P_\omega, \pi_\theta}[R_t \mid s_t = s, a_t = a], \quad q^*(s, a, \omega) = \mathbb{E}_{P_\omega, \pi^*}[R_t \mid s_t = s, a_t = a],$$

where $\mathbb{E}$ is the expectation. Note that we introduce $\omega$ to the argument to explain the $Q$-value dependence on $\omega$. Lastly, we define the value function as

$$v^{\pi_\theta}(s, \omega) = \mathbb{E}_{P_\omega, \pi_\theta}[R_t \mid s_t = s], \quad v^*(s, \omega) = \mathbb{E}_{P_\omega, \pi^*}[R_t \mid s_t = s].$$

In the following, we will drop the $\omega$ subscript for simplicity and define the expectile (optimal) value function, that follows the recursive Bellman equation

$$v^{\pi, P}(s) = v^\pi(s) = \mathbb{E}_{a \sim \pi(\cdot | s)}[\underbrace{r(s, a) + \gamma \mathbb{E}_{P_{sa}}[v^\pi]}_{\triangleq q^\pi(s,a)}], \quad v^*(s) = \max_{a \in \mathcal{A}}(\underbrace{r(s, a) + \gamma \mathbb{E}_{P_{sa}}[v^*]}_{\triangleq q^*(s,a)})). \quad (1)$$

Finally, we define the classical Bellman Operator and optimal Bellman Operator that are $\gamma$-contractions, so iteration of these operators leads $v^\pi$ or $v^*$:

$$(T^\pi v)(s) := (T^\pi_{r,P} v)(s) = \sum_a \pi(a|s)(r(s, a) + \gamma \mathbb{E}_{P_{sa}}[v]) \quad (2)$$

$$(T^* v)(s) := (T^{\pi^*}_{r,P} v)(s) = \max_a(r(s, a) + \gamma \mathbb{E}_{P_{sa}}[v]). \quad (3)$$

### 3.2 ROBUST MDPS

Once classical MDPs are defined, we can define robust (optimal) Bellman operators $\mathcal{T}^\pi_{\mathcal{U}}$ and $\mathcal{T}^*_{\mathcal{U}}$,

$$(T^\pi_{\mathcal{U}} v)(s) := \min_{r, P \in \mathcal{U}} (T^\pi_{r,P} v)(s), (T^*_{\mathcal{U}} v)(s) := \max_{\pi \in \Delta_A} \min_{r, P \in \mathcal{U}} (T^\pi_{r,P} v)(s), \quad (4)$$

where $P$ and $r$ belong to the uncertainty set $\mathcal{U}$. The optimal robust Bellman operator $T^*_{\mathcal{U}}$ and robust Bellman operator $T^\pi_{\mathcal{U}}$ are $\gamma$-contraction maps for any policy $\pi$ (Iyengar, 2005, Thm. 3.2) if the uncertainty set $\mathcal{U}$ is a subset of $\Delta_s$ where $\Delta_s$ it the simplex of $|S|$ elements so that the transition kernel is valid. Finally, for any initial values $v_0^\pi, v_0^*$, sequences defined as $v_{n+1}^\pi := T^\pi_{\mathcal{U}} v_n^\pi$ and $v_{n+1}^* := T^*_{\mathcal{U}} v_n^*$ converge linearly to their respective fixed points, that is $v_n^\pi \to v_{\mathcal{U}}^\pi$ and $v_n^* \to v_{\mathcal{U}}^*$.

## 3.3 EXPECTILES

Let's first define expectiles. For $\alpha \in (0, 1)$ and $X$ a random variable, the $\alpha$-expectile is defined as $m_\alpha(X) = \operatorname{argmin}_m \mathbb{E}_x[L_2^\alpha(x - m)]$ with $L_2^\alpha(u) = |\alpha - \mathbb{1}_{\{u<0\}}|u^2 = \alpha u_+^2 + (1 - \alpha)u_-^2$, where $u_+ = \max(u, 0)$ and $u_- = \max(-u, 0)$. We can recover the classical mean with $m_{\frac{1}{2}}(X) = \mathbb{E}[X]$ as $L_2^{1/2}(u) = u^2$. Expectiles are gaining interest in statistics and finance as they induce the only law-invariant, coherent (Artzner et al., 1999) and elicitable (Gneiting, 2011) risk measure. Using the coherent property representation (Delbaen, 2000), one has that $\rho : L^\infty \to \mathbb{R}$ is a coherent risk measure if and only if there exists a closed convex set $\mathcal{P}$ of $P$-absolutely continuous probability measures such that $\rho(X) = \inf_{Q \in \mathcal{P}} \mathbb{E}_Q[X], \forall X \in L^\infty$. with $L^\infty$ the vector space of essentially bounded measurable functions with the essential supremum norm. The uncertainty set induced by expectiles as been described by Delbaen (2013) as $m_\alpha(X) = \min_{Q \in \mathcal{E}} \mathbb{E}_Q[X]$ such as

$$\mathcal{E} = \left\{ Q \in \mathcal{P} \mid \exists \eta > 0, \eta\sqrt{\frac{\alpha}{1 - \alpha}} \le \frac{dQ}{dP} \le \sqrt{\frac{(1 - \alpha)}{\alpha}}\eta \right\} \tag{5}$$

where we define $\frac{dQ}{dP}$ as the Radon-Nikodym derivative of $Q$ with respect to $P$. Here, the uncertainty set corresponds thus to a lower and upper bound on $\frac{dQ}{dP}$ with a quantity depending on the degree of uncertainty. For $\alpha = 1/2$, the uncertainty set becomes the null set and we retrieve the classical mean. This variational form of the expectile will be useful to link risk-sensitive and robust MDPs formulation in the next section.

## 4 EXPECTRL METHOD

First, we introduce the Expectile Bellman Operator and then we will explain our proposed method `ExpectRL` and `AutoExpectRL` that work both in classic and robust cases.

### 4.1 EXPECTILE BELLMAN OPERATOR

In this section, we derive the loss and explain our approach. Recall that for $\alpha \in (0, 1)$ and $X$ a random variable taking value $x$ and following a probability law $P$, the $\alpha$-expectile is denoted $m_\alpha(X)$ or $m_\alpha(P, x)$ in the following. Writing the classical Bellman operator for $q$ function

$$(Tq)(s, a) = r(s, a) + \gamma\langle P_{sa}, v\rangle = r(s, a) + \gamma\mathbb{E}_{s' \sim P_{sa}(\cdot)}[v(s')].$$

and denoting $V_{sa}$ the random variable which is equal to $v(s')$ with probability $P_{sa}(s')$, it holds that:

$$(Tq)(s, a) = r(s, a) + \gamma m_{\frac{1}{2}}(V_{sa}) = r(s, a) + \gamma m_{\frac{1}{2}}(P_{sa}, v).$$

Our method consists instead in considering the following Expectile Bellman operator

$$(T_\alpha q)(s, a) = r(s, a) + \gamma m_\alpha(V_{sa}). \tag{6}$$

With $\alpha < \frac{1}{2}$, Eq. (6) allows to learn a robust policy, in the sense that it is a pessimistic estimate about the value we bootstrap according to the value sampled according to the nominal kernel. Next, we define the expectile value of a given policy and the optimal expectile value as:

$$v_\alpha^\pi(s) = \mathbb{E}_{a \sim \pi(\cdot|s)}[\underbrace{r(s, a) + \gamma m_\alpha(P_{sa}, v_\alpha^\pi)}_{\triangleq q_\alpha^\pi(s, a)}], \qquad v_\alpha^*(s) = \max_{a \in \mathcal{A}}(\underbrace{r(s, a) + \gamma m_\alpha(P_{sa}, v_\alpha^*)}_{\triangleq q_\alpha^*(s, a)}). \tag{7}$$

With $\alpha = \frac{1}{2}$, we retrieve the standard Bellman equations but we consider $\alpha < \frac{1}{2}$ for the robust case. Finally, we define (optimal) expectile Bellman Operator as:

$$(T_\alpha^\pi v)(s) = \sum_a \pi(a|s)(r(s, a) + \gamma m_\alpha(P_{sa}, v)). \quad (T_\alpha^* v)(s) = \max_a(r(s, a) + \gamma m_\alpha(P_{sa}, v)).$$

**Theorem 4.1.** *The (optimal) Expectile Bellman Operators are $\gamma$-contractions for the sup norm. (proof in Appx. B).*

So as $T_\alpha^\pi$ and $T_\alpha^*$ are $\gamma$-contractions, it justifies the definition of fixed point $v_\tau^\pi$ and $v_\alpha^*$. The central idea to show that expectile bootstrapping or `ExpectRL` is implicitly equivalent to Robust RL comes (Zhang et al., 2023) where we try to estimate the optimal robust value function $v_\mathcal{E}^* = \max_\pi \min_{Q \in \mathcal{E}} v^{\pi, Q}$.

**Theorem 4.2.** *The (optimal) Expectile value function is equal to the (optimal) robust value function*

$$v_\alpha^*(s) = v_\mathcal{E}^\pi := \max_\pi \min_{Q \in \mathcal{E}} v^{\pi,Q}, \quad v_\alpha^\pi(s) = v_\mathcal{E}^\pi := \min_{Q \in \mathcal{E}} v^{\pi,Q} \tag{8}$$

where $\mathcal{E}$ is defined in 3.3. Proof can be found in B.2. Note that his formulation does not converge to the expectile of the value distribution but to $v_\mathcal{E}^*$ the robust value function. Moreoever, for $\alpha > 1/2$, Expectile Bellman operator is not anymore a contraction and there is no theoretical convergence guarantees for risk-seeking RL, not considered here. Now that expectile operators are defined, we will define the related loss.

### 4.2 THE EXPECRL LOSS

In this section, we present the method more from a computational and practical point of view. As stated before, this method can be plugged into any RL algorithm where a $Q$-function is estimated, which included any $Q$-function-based algorithm or some actor-critic framework during the critic learning. For a given algorithm, the only modification relies on modifying the $L_2$ loss in the $Q$-value step by the Expectile loss. Given a target $y(r, s') = r + \gamma Q_{\phi, \text{targ}}(s', \pi(s'))$ with reward $r$, policy $\pi$, we propose to minimize

$$L(\phi, \mathcal{D}) = \mathbb{E}_{(s,a,r,s') \sim \mathcal{D}} [L_2^\alpha (Q_\phi(s, a) - y(r, s'))], \tag{9}$$

where $L_2^\alpha$ is the expectile loss defined in Section 3.3. For $\alpha = 1/2$, the expectile coincides with the classical mean, and we retrieve the classical $L_2$ loss present in most RL algorithms. We will use TD3 as a baseline and replace the learning of the critic with this loss. The actor loss remains the same in the learning process. With ExpectRL, only one critic is needed, replacing the double critic present in this algorithm. We will compare our method with the classical TD3 algorithm using the twin critic trick and TD3 with one critic to see the influence of our method.

### 4.3 EXPECRL METHOD WITH DOMAIN RANDOMISATION

From a practical point of view, many Robust RL algorithms such as M2TD3 (Tanabe et al., 2022), M3DDPG (Li et al., 2019), and RARL (Pinto et al., 2017) not only interact with the nominal environment but also with environments that belong to the uncertainty set $\mathcal{U}$. Sampling trajectories from the entire uncertainty set allows algorithms to get knowledge from dangerous trajectories and allows algorithms to generalize better than algorithms that only sample from the nominal. Receiving information about all environments that need to be robust during the training phase, the algorithm tends to obtain better performance on minimum performance over these environments on testing. With the same idea of generalization, Domain Randomisation (DR) (Tobin et al., 2017) focuses not on the worst case under the uncertainty set but on the expectation. Given a point of the uncertainty set $P_\omega \in \mathcal{U}$, the DR objective is: $\pi_{\text{DR}}^* = \text{argmax}_\pi \mathbb{E}_{\omega \in \Omega, s \sim P_\omega^0}[v^\pi(s, w)]$. In other words, DR tries to find the best policy on average over all environments in the uncertainty set. The approach we propose to be competitive on a robust benchmark is to find the best policy using ExpectRL under domain randomization or

$$\pi_{\text{DR},\alpha}^* = \underset{\pi}{\text{argmax}} \, \mathbb{E}_{\omega \in \Omega, s \sim P_\omega^0}[v_\alpha^\pi(s, \omega)] = \underset{\pi}{\text{argmax}} \, \mathbb{E}_{\omega \in \Omega, s \sim P_\omega^0}[\min_{P_\omega \in \mathcal{E}} v^{\pi,P}(s, \omega)], \tag{10}$$

where $v_\alpha^\pi(s, \omega)$ is the expectile value function under uncertainty kernel $P_\omega$ and $\mathcal{E}$ defined in Section 3.3. Using this approach, we hope to get sufficient information from all the environments using DR and improve robustness and worse-case performance using ExpectRL. The advantage of the approach is that any algorithm can be used for learning the policy, sampling from the entire uncertainty set uniformly and replacing the critic loss of this algorithm learning with ExpecRl loss. The effectiveness of this algorithm on a Robust benchmark will be conducted in Section 3. Getting an algorithm that is mathematically founded and which tries to get the worst-case performance, the last question is how to choose the degree of pessimism or $\alpha \in (0, 1/2)$ in practice. The following section tries to answer this question using a bandit algorithm to auto-tune $\alpha$.

### 4.4 AUTO-TUNING OF THE EXPECTILE $\alpha$ USING BANDIT

In the context of varying levels of uncertainty across environments, the selection of an appropriate expectile $\alpha$ becomes contingent on the specific characteristics of each environment. To automate

| | TD3 Twin Critic | TD3 1 critic | ExpectRL best Expectile | AutoExpectRL |
|---|---|---|---|---|
| $Ant(\times 10^3)$ | $3.65 \pm 0.33$ | $1.90 \pm 0.07$ | $\mathbf{4.46 \pm 0.12}$ | $4.27 \pm 0.25$ |
| $HalfCheetah(\times 10^3)$ | $\mathbf{10.91 \pm 0.14}$ | $10.36 \pm 0.54$ | $10.42 \pm 0.13$ | $10.40 \pm 0.09$ |
| $Hopper(\times 10^3)$ | $2.88 \pm 0.10$ | $2.022 \pm 0.09$ | $\mathbf{3.10 \pm 0.05}$ | $3.03 \pm 0.11$ |
| $Walker(\times 10^3)$ | $2.95 \pm 0.12$ | $2.35 \pm 0.25$ | $\mathbf{3.22 \pm 0.11}$ | $3.02 \pm 0.09$ |
| $HumanoidStandup(\times 10^5)$ | $1.101 \pm 0.09$ | $1.087 \pm 0.09$ | $\mathbf{1.197 \pm 0.05}$ | $1.143 \pm 0.010$ |

Table 1: Expectile vs Twin-critic, Mean performance $\pm$ standard error, on 10 train seed

the process of choosing the optimal expectile, we employ a bandit algorithm, specifically the Exponentially Weighted Average Forecasting algorithm (Cesa-Bianchi & Lugosi, 2006). We denote this method as `AutoExpectRL`. This formulation adopts the multi-armed bandit problem, where each bandit arm corresponds to a distinct value of $\alpha$. We consider a set of $D$ expectiles making predictions from a discrete set of values $\{\alpha_d\}_{d=1}^D$. At each episode $m$, a cumulative reward $R_m$ is sampled, and a distribution over arms $\mathbf{p}_m \in \Delta_D$ is formed, where $\mathbf{p}_m(d) \propto \exp(w_m(d))$. The feedback signal $f_m \in \mathbb{R}$ is determined based on the arm selection as the improvement in performance, specifically $f_m = R_m - R_{m-1}$, where $R_m$ denotes the cumulative reward obtained in the episode $m$. Then, $w_{m+1}$ is obtained from $w_m$ by modifying only the $d_m$ according to $w_{m+1}(d_m) = w_m(d_m) + \eta \frac{f_m}{\mathbf{p}_m(d)}$ where $\eta > 0$ is a step size parameter. The exponential weights distribution over $\alpha$ values at episode $m$ is denoted as $\mathbf{p}_m^\alpha$. This approach can be seen as a form of model selection akin to the methodology presented by Pacchiano et al. (2020). Notably, instead of training distinct critics and actors for each $\alpha$ choice, our approach updates one single neural network for the critic and one single neural network for the actor. In both critic and actor, neural networks are composed of one common body and different heads for every value of $\alpha$, in our case 4 values for $\{\alpha_d\}_{d=1}^D = \{0.2, 0.3, 0.4, 0.5\}$. The critic's heads correspond to the 4 expectile losses for different values of $\alpha$. The actor's neural network is trained using 4 classical TD3 losses, evaluated with action chosen by one specific head of the actor. Then in both critic and actor, the 4 losses are summed, allowing an update of all heads at each iteration. Finally, the sampling of new trajectories is done using the chosen head of the actor, proposed by the bandit algorithm. More details about implementation can be found in Appendix C. Intuitively, when the agent receives a higher reward compared to the previous trajectory, the probability of choosing this arm is increased to encourage this arm to be picked again. Note that the use of a bandit algorithm to automatically select hyperparameters in an RL algorithm has been proposed in other contexts, such as Moskovitz et al. (2021); Badia et al. (2020). The `AutoExpectRL` method allows picking automatically expectile $\alpha$ and reduces hyperparameter tuning. Practical details can be found in Appendix C where we expose the neural network architecture of this problem and associated losses. Note that this approach does not work in the DR setting as uncertainty parameters change between trajectories in DR. It is difficult for the algorithm to know if high or low rewards on trajectories come because the uncertainty parameter leads to small rewards, or if it is due to bad expectile picked at this iteration.

## 5 EMPIRICAL RESULT ON MUJOCO

The Mujoco benchmark is employed in this experiment due to its significance for evaluating robustness in the context of continuous environments, where physical parameters may vary. In contrast, the Atari benchmark very deterministic with discrete action space without physical parameters cannot change during the testing period. In this section, we compare the performance of the TD3 algorithm using the twin critic method during learning, only one critic, and finally our method `ExpectRL`. The different values of $\alpha$ are $\{\alpha_d\}_{d=1}^D = \{0.2, 0.3, 0.4, 0.5\}$. We can notice that `ExpectRL` with $\alpha = 0.5$ is exactly TD3 with one critic. Here, we only interact with the nominal and there is no notion of robustness. The mean and standard deviation are reported in Table 1, where we use 10 seeds of 3M steps for training, each evaluated on 30 trajectories. The last column is our last algorithm, `AutoExpectRL`. In all environments except HalfCheetah, `ExpectRL` with fine-tuning of $\alpha$ has the best score and `AutoExpectRL` has generally close results. The scores for every expectiles can be found in AppendixE. In Halcheetah 1 environment, it seems that no pessimism about $Q$-function is needed and our method `ExpectRL` is outperformed by TD3 with twin critic. Similar observations have been observed in Moskovitz et al. (2021) on this environment. Moreover, results for $\alpha = 0.5$ and $\alpha = 0.4$ are very close in Appendix E while the variance is reduced using $\alpha = 0.4$. Results of Table 1 show that it is possible to replace the twin critic approach with only one critic with the relevant value of pessimism or expectile. Moreover, one can remark

| | TD3 mean | ExpectRL mean | Auto mean | TD3 worst | ExpectRL worst | Auto worst |
|---|---|---|---|---|---|---|
| $Ant1$ | $2.76 \pm 0.5$ | $3.55 \pm 0.65$ | $\mathbf{3.55 \pm 0.51}$ | $2.22 \pm 0.5$ | $2.65 \pm 0.57$ | $\mathbf{2.71 \pm 0.43}$ |
| $Ant2$ | $2.28 \pm 0.09$ | $\mathbf{2.50 \pm 0.89}$ | $2.41 \pm 0.77$ | $1.59 \pm 0.08$ | $\mathbf{2.49 \pm 0.94}$ | $2.42 \pm 0.51$ |
| $Ant3$ | $0.31 \pm 1.13$ | $\mathbf{0.54 \pm 0.08}$ | $0.53 \pm 0.69$ | $-0.99 \pm 1.13$ | $-0.94 \pm 0.21$ | $\mathbf{-0.88 \pm 0.34}$ |
| $Half1$ | $2.79 \pm 0.22$ | $\mathbf{3.05 \pm 0.48}$ | $2.98 \pm 0.19$ | $-0.34 \pm 0.04$ | $\mathbf{-0.27 \pm 0.19}$ | $-0.27 \pm 0.21$ |
| $Half2$ | $\mathbf{2.63 \pm 0.20}$ | $2.51 \pm 0.41$ | $2.58 \pm 0.32$ | $-0.53 \pm 0.06$ | $\mathbf{-0.223 \pm 0.16}$ | $-0.23 \pm 0.10$ |
| $Half3$ | $\mathbf{2.47 \pm 0.18}$ | $2.45 \pm 0.42$ | $2.39 \pm 0.15$ | $-0.61 \pm 0.08$ | $\mathbf{-0.557 \pm 0.27}$ | $-0.58 \pm 0.09$ |
| $Hopper1$ | $2.39 \pm 0.14$ | $\mathbf{2.76 \pm 0.04}$ | $2.52 \pm 0.11$ | $0.4 \pm 0.02$ | $0.44 \pm 0.01$ | $\mathbf{0.449 \pm 0.15}$ |
| $Hopper2$ | $1.54 \pm 0.17$ | $\mathbf{2.06 \pm 0.01}$ | $1.87 \pm 0.02$ | $0.21 \pm 0.04$ | $\mathbf{0.32 \pm 0.03}$ | $0.32 \pm 0.03$ |
| $Hopper3$ | $1.15 \pm 0.14$ | $\mathbf{1.43 \pm 0.02}$ | $1.433 \pm 0.09$ | $0.14 \pm 0.03$ | $\mathbf{0.25 \pm 0.22}$ | $0.242 \pm 0.19$ |
| $Walker1$ | $3.12 \pm 0.2$ | $\mathbf{3.66 \pm 0.68}$ | $3.58 \pm 0.27$ | $0.68 \pm 0.12$ | $\mathbf{2.77 \pm 0.15}$ | $1.99 \pm 0.13$ |
| $Walker2$ | $2.70 \pm 0.2$ | $\mathbf{3.98 \pm 0.58}$ | $3.88 \pm 0.61$ | $0.28 \pm 0.07$ | $\mathbf{1.36 \pm 0.82}$ | $1.11 \pm 0.15$ |
| $Walker3$ | $2.60 \pm 0.18$ | $\mathbf{3.84 \pm 0.45}$ | $3.58 \pm 0.15$ | $0.17 \pm 0.06$ | $0.65 \pm 0.12$ | $\mathbf{0.87 \pm 0.09}$ |
| $Humanoid1$ | $1.03 \pm 0.4$ | $1.12 \pm 0.25$ | $\mathbf{1.13 \pm 0.26}$ | $0.85 \pm 0.07$ | $\mathbf{0.97 \pm 0.23}$ | $0.98 \pm 0.24$ |
| $Humanoid2$ | $1.03 \pm 0.3$ | $\mathbf{1.13 \pm 0.15}$ | $1.11 \pm 0.12$ | $0.73 \pm 0.07$ | $\mathbf{0.83 \pm 0.23}$ | $0.80 \pm 0.18$ |
| $Humanoid3$ | $1.01 \pm 0.3$ | $\mathbf{1.06 \pm 0.13}$ | $1.05 \pm 0.18$ | $0.57 \pm 0.04$ | $\mathbf{0.71 \pm 0.21}$ | $0.68 \pm 0.09$ |

Table 2: Result on Robust Benchmark for TD3 `ExpectRL` and `AutoExpectRL`. Results are $\times 10^3$ bigger for all environments except for Humanoid where results are $\times 10^5$ bigger.

| | DR+ExpectRL(m) | M2TD3(m) | DR(m) | DR+ExpectRL(w) | M2TD3(w) | DR(w) |
|---|---|---|---|---|---|---|
| $Ant1$ | $4.84 \pm 0.43$ | $4.51 \pm 0.08$ | $\mathbf{5.25 \pm 0.1}$ | $3.36 \pm 0.55$ | $\mathbf{3.84 \pm 0.1}$ | $3.51 \pm 0.08$ |
| $Ant2$ | $5.63 \pm 0.43$ | $5.44 \pm 0.05$ | $\mathbf{6.32 \pm 0.09}$ | $2.72 \pm 0.42$ | $\mathbf{4.13 \pm 0.11}$ | $1.64 \pm 0.13$ |
| $Ant3$ | $2.86 \pm 1.03$ | $2.66 \pm 0.22$ | $\mathbf{3.62 \pm 0.11}$ | $\mathbf{0.28 \pm 0.35}$ | $0.10 \pm 0.10$ | $-0.32 \pm 0.03$ |
| $Half1$ | $5.3 \pm 0.59$ | $3.89 \pm 0.06$ | $\mathbf{5.93 \pm 0.18}$ | $2.86 \pm 0.99$ | $3.14 \pm 0.10$ | $\mathbf{3.19 \pm 0.08}$ |
| $Half2$ | $5.25 \pm 0.32$ | $4.35 \pm 0.05$ | $\mathbf{5.79 \pm 0.15}$ | $1.77 \pm 0.31$ | $\mathbf{2.61 \pm 0.16}$ | $2.12 \pm 0.13$ |
| $Half3$ | $4.52 \pm 0.24$ | $3.79 \pm 0.09$ | $\mathbf{5.54 \pm 0.16}$ | $1.02 \pm 0.24$ | $0.93 \pm 0.21$ | $\mathbf{1.09 \pm 0.06}$ |
| $Hopper1$ | $2.58 \pm 0.23$ | $\mathbf{2.68 \pm 0.11}$ | $2.57 \pm 0.15$ | $\mathbf{0.64 \pm 0.20}$ | $0.62 \pm 0.45$ | $0.53 \pm 0.26$ |
| $Hopper2$ | $\mathbf{2.53 \pm 0.22}$ | $2.51 \pm 0.07$ | $1.89 \pm 0.08$ | $\mathbf{0.55 \pm 0.20}$ | $0.53 \pm 0.28$ | $0.47 \pm 0.02$ |
| $Hopper3$ | $\mathbf{2.21 \pm 0.33}$ | $0.85 \pm 0.07$ | $1.5 \pm 0.07$ | $\mathbf{0.39 \pm 0.07}$ | $0.28 \pm 0.25$ | $0.21 \pm 0.03$ |
| $Walker1$ | $\mathbf{3.77 \pm 0.89}$ | $3.70 \pm 0.31$ | $3.59 \pm 0.26$ | $\mathbf{3.41 \pm 0.05}$ | $2.83 \pm 0.39$ | $2.19 \pm 0.42$ |
| $Walker2$ | $\mathbf{4.75 \pm 0.57}$ | $4.72 \pm 0.12$ | $4.54 \pm 0.31$ | $2.74 \pm 0.61$ | $\mathbf{3.14 \pm 0.39}$ | $2.31 \pm 0.51$ |
| $Walker3$ | $4.39 \pm 0.37$ | $4.27 \pm 0.21$ | $\mathbf{4.48 \pm 0.16}$ | $1.14 \pm 0.79$ | $\mathbf{1.34 \pm 0.43}$ | $1.32 \pm 0.34$ |
| $Humanoid1$ | $\mathbf{1.21 \pm 0.23}$ | $1.08 \pm 0.04$ | $1.12 \pm 0.05$ | $\mathbf{1.04 \pm 0.86}$ | $0.93 \pm 0.07$ | $0.96 \pm 0.06$ |
| $Humanoid2$ | $\mathbf{1.23 \pm 0.22}$ | $0.97 \pm 0.04$ | $1.06 \pm 0.04$ | $\mathbf{0.86 \pm 0.28}$ | $0.65 \pm 0.07$ | $0.73 \pm 0.78$ |
| $Humanoid3$ | $\mathbf{1.12 \pm 0.35}$ | $1.09 \pm 0.06$ | $1.04 \pm 0.07$ | $\mathbf{0.84 \pm 0.26}$ | $0.62 \pm 0.06$ | $0.54 \pm 0.34$ |

Table 3: Result on Robust Benchmark for `ExpectRL` + DR , M2TD3 and DR. Results are $\times 10^3$ bigger for all environments except for Humanoid results are $\times 10^5$ bigger. The mean performance is denoted $(m)$ and worst case $(w)$.

in Appendix E that in Hopper, Walker, and Ant environment, high pessimism is needed to get an accurate $Q$ function and better results, with a value of $\alpha = 0.2$ or $\alpha = 0.3$ whereas less pessimism with $\alpha = 0.4$ is needed for HumanoidStandup and HalfCheetah. Note that the value of $\alpha = 0.5$ is never chosen and leads to generally the worst performance as reported in column TD3 with one critic which coincides with $\alpha = 0.5$. Finally, the variance is also decreased using our method compared to TD3 with twin critics or TD3 with one critic. Finally, our method `AutoExectRL` allows choosing automatically the expectile almost without loss of performance and outperforming TD3, except on the environment HalfCheetah. Learning curves can be found in Appendix E.

## 6 EMPIRICAL RESULTS ON ROBUST BENCHMARK

This section presents an assessment of the worst-case and average performance and generalization capabilities of the proposed algorithm. The experimental validation was conducted on optimal control problems utilizing the MuJoCo simulation environments (Todorov et al., 2012). The performance of the algorithm was systematically benchmarked against state-of-the-art robust RL M2TD3 as it is state of the art compared to other algorithms methodologies, M3DDPG, and RARL. Furthermore, a comparative analysis was undertaken with Domain Randomization (DR) as introduced by Tobin et al. (2017) for a comprehensive evaluation. To assess the worst-case performance of the policy $\pi$ under varying uncertainty parameters $\omega \in \Omega$, following the benchmark of Tanabe et al. (2022) or Zouitine et al. (2024), 30 evaluations of the cumulative reward were conducted for each uncertainty parameter value $\omega_1, \ldots, \omega_K \in \Omega$. Specifically, $R_k(\pi)$ denotes the cumulative reward on $\omega_k$, averaged over 30 trials. Subsequently, $R_{\text{worst}}(\pi) = \min_{1 \leqslant k \leqslant K} R_k(\pi)$ (denoted (w) in Table 2 and 3) was computed as an estimate of the worst-case performance of $\pi$ on $\Omega$. Additionally, the average performance was computed as $R_{\text{average}}(\pi) = \frac{1}{K} \sum_{k=1}^{K} R_k(\pi)$ (denoted (m) in Table 2 and 3). For the evaluation process, $K$ uncertainty parameters $\omega_1, \ldots, \omega_K$ were chosen according to the dimensionality of $\omega$:

for 1D $\omega$, $K = 10$ equally spaced points on the 1D interval $\Omega$; for 2D $\omega$, 10 equally spaced points were chosen in each dimension of $\Omega$, resulting in $K = 100$ points; and for 3D $\omega$, 10 equally spaced points were selected in each dimension of $\Omega$, resulting in $K = 1000$ points or different environments. Each approach underwent policy training 10 times in each environment. The training time steps $T_{\max}$ were configured as $2M, 4M$, and $5M$ for scenarios with $1D, 2D$, and $3D$ uncertainty parameters respectively, following Tanabe et al. (2022). Table 6 summarizes the different changes of parameters in the environments. The final policies obtained from training were then evaluated for their worst-case performances and average performance over all uncertainty parameters. The results are the following.

We first demonstrate that our method `ExpectRl` is more robust than the classical RL algorithm. To do so, we conduct the benchmark task presented previously on TD3 algorithm (with twin critic trick) as a baseline and our method `ExpectRl`. As exposed in Table 2, our method outperforms TD3 in all environments on worst-case performance, which was expected as TD3 is not designed by nature to be robust and to maximize a worst-case performance. Moreover, `AutoExpectRL` has good and similar performance compared to the best expectile like in Table 1. As TD3 has sometimes very bad performance, our method also performs better on average over all environments except HalfCheetah 2 and HalfCheetah 3. These two environments required more exploration, and pessimism is in general not a good thing for these tasks. Moreover, robustness is not needed in HalfCheetah environments that are already quite stable compared to other tasks in Mujoco. However, `ExpectRL` needs to be compared with algorithms designed to be robust, such as M2TD3 which has state-of-the-art performance on this benchmark.

If performance of `ExpectRL` in Table 2 and the performance of M2TD3 in Table 3 are compared, we can observe a large difference on many tasks where M2TD3 outperforms, in general, our method. This is because sampling trajectories from the entire uncertainty set allows M2TD3 to get knowledge from dangerous trajectories and allows the algorithm to generalize better than our method, which only samples from the nominal. The comparison between methods is then not fair for `ExpectRl` which has only access to samples from the nominal and this is why the method `ExpectRL + DR` was introduced. Receiving information about all environments that need to be robust during the training phase, the algorithm tends to obtain better performance on minimum performance over these environments on testing. Table 3 shows the result on average and on worst-case performance between our second method `ExpectRL` + DR with tuning of $\alpha$ against M2TD3 and DR approach. Recall that `AutoExpectRL` cannot be used with DR as mentioned at the end of Section 4.4.

In terms of worst-case performance, our method outperforms 9 times M2TD3 (8 times in bold and one time when DR is better in general for HalfCheetaht3) and has a worse performance on 6 tasks compared to M2TD3. Our method is therefore competitive with the state of the art in robust algorithms such as M2TD3, which already outperformed M3DDPG and RARL on worst-case performance. Except on Hopper1, our method outperforms M2TD3 on average, results which show that M2TD3 is very pessimistic compared to our method. However, in terms of average results, we can see that DR, which is designed to be good on average across all environments, generally performs better than our method and M2TD3 expect on Hopper, Walker1 and 2, and HumanoidStandup which are not stable and need to be robustified to avoid catastrophic performance that affect too much the mean performance over all environment. Moreover, compared to M2TD3, our method `ExpecRL`, even without auto fine-tuning of $\alpha$, has the advantage of having fewer parameter tuning compared to the M2TD3 algorithm.

## 7 CONCLUSION AND PERSPECTIVES

We propose a simple method, `ExpectRL` that replaces the classic loss $L_2$ of the critic with an expectile loss. Moreover, we show that it can also lead to a Robust RL algorithm and demonstrate the effectiveness of our method combined with DR on a robust RL Benchmark. The limitations of our method are that `AutoExpectRL` allows fine-tuning of $\alpha$ only without combining with DR. Another limitation is that the uncertainty set defined with expectile is not fully interpretable, and other algorithms with a more interpretable set and similar experimental results would be interesting. About future perspectives, we demonstrate the effectiveness of our method using as baselines TD3, but our method can be easily adapted to any algorithm using a $Q$-function such as classical DQN, SAC, and other algorithms both with discrete or continuous action space.

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

## A   APPENDIX

## B   PROOF

**Theorem B.1.**

$$(T_\alpha^\pi v)(s) = \sum_a \pi(a|s)(r(s,a) + \gamma m_\alpha(P_{sa}, v)). \tag{11}$$

*this is a contraction:*

The expectile satisfies the following properties (Bellini et al., 2014; Bellini & Di Bernardino, 2017):

1. Translation invariance: $m_\tau(X + h) = m_\alpha(X) + h$
2. Monotonicity: $X \leq Y$ a.s. $\Rightarrow m_\tau(X) \leq m_\alpha(Y)$
3. Positive homogeneity:
   $\lambda \geq 0 \Rightarrow m_\alpha(\lambda X) = \lambda m_\tau(X)$
4. Superadditivity, for $\alpha \leq \frac{1}{2}, m_\tau(X + Y) \geq m_\alpha(X) + m_\alpha(Y)$.

So,

$$(T_\alpha^\pi v_1)(s) - (T_\alpha^\pi v_2)(s) = \sum_a \pi(a|s)(r(s,a) + \gamma m_\alpha(P_{sa}, v_1)) \tag{12}$$

$$- \sum_a \pi(a|s)(r(s,a) + \gamma m_\alpha(P_{sa}, v_2)) \tag{13}$$

$$= \gamma \sum_a \pi(a|s)(m_\alpha(P_{sa}, v_1) - m_\alpha(P_{sa}, v_2)) \tag{14}$$

$$\leq \gamma \sum_a \pi(a|s)(m_\alpha(P_{sa}, v_2 + \|v_2 - v_1\|_\infty) - m_\alpha(P_{sa}, v_2)) \tag{15}$$

(by monotonicity) and $v_1 \leq v_2 + \|v_2 - v_1\|_\infty)$

$$= \gamma \sum_a \pi(a|s)(m_\alpha(P_{sa}, v_2) + \|v_2 - v_1\|_\infty - m_\alpha(P_{sa}, v_2)) \text{ (by translation invariance)} \tag{16}$$

$$= \gamma \|v_2 - v_1\|_\infty. \tag{17}$$

In the same manner, $T_\alpha^*$ is also a contraction, as the only line of this proof that differs is replacing the expectation by a $\max_a$. As maximum operator 1-Lipschitz, (ie) $\max_a f(a) - \max g(a) \leq \max f(a) - g(a)$, we obtain $\gamma$- contraction results also for the optimal Bellman operator $T_\alpha^*$.

Similar ideas exist in Zhang et al. (2023), which show similar properties for risk-sensitive MDPs defined through a convex risk measure, even though they do not consider explicitly the expectile which is a convex risk measure for $\alpha < 1/2$.

**Theorem B.2.** *The (optimal) Expectile value function is equal to the (optimal) robust value function*

$$v_\alpha^*(s) = v_{\mathcal{E}}^\pi := \max_\pi \min_{Q \in \mathcal{E}} v^{\pi, Q} \tag{18}$$

$$v_\alpha^\pi(s) = v_{\mathcal{E}}^\pi := \min_{Q \in \mathcal{E}} v^{\pi, Q} \tag{19}$$

where $\mathcal{E}$ is defined in section 3.3 or below.

*Proof.* This theorem is just an adaptation of Theorem 2 in Zhang et al. (2023) where we use expectile risk measure $m_\alpha(X)$ which implicitly defined the uncertainty set for robust $\mathcal{E}$ such that :

$$m_\alpha(X) = \min_{Q \in \mathcal{E}} \mathbb{E}_Q[X];$$

$$\mathcal{E} = \left\{ Q \in \mathcal{P} \mid \exists \eta > 0, \sqrt{\frac{\alpha}{1 - \alpha}}\eta \leq \frac{dQ}{dP} \leq \sqrt{\frac{(1 - \alpha)}{\alpha}}\eta \right\}$$

where $\mathcal{P}$ is the set of $P$-absolutely continuous probability measures. In Theorem Zhang et al. (2023), they link Risk sensitive MDPs (in our case expectile formulation) with Regularised Robust MDPs. In our case, we can rewrite the classical RMDPs to Regularised-Robust MDPs such that:

$$v_{\mathcal{E}}^* = \max_\pi \min_{Q \in \mathcal{E}} v^{\pi, Q} = \max_\pi \min_{Q \in \mathcal{E}} \mathbb{E}\Big[\sum_t \gamma^t r(s_t, a_t)\Big]$$

$$= \max_\pi \min_{Q \in \mathcal{P}} \mathbb{E}\Big[\sum_t \gamma^t (r(s_t, a_t) + \gamma D(P_{t;s_t,a_t}, Q_{t;s_t,a_t})\Big]$$

with $D$ a penalty function that can be chosen as KL diverengence for example and $P_{t;s_t,a_t}$ the transition kernel at time $t$ with current state action $(s_t, a_t)$. For the expectile risk measure, the corresponding $D$ is simply:

$$D(P, Q) = \begin{cases} 0 & \text{if } \eta\sqrt{\frac{\alpha}{1-\alpha}} \leq P(s)/Q(s) \leq \sqrt{\frac{(1-\alpha)}{\alpha}}\eta, \forall s \in S \\ +\infty & \text{otherwise.} \end{cases}$$

where $\eta$ is defined in 3.3. Using Theorem 2 of Zhang et al. (2023), we have directly that :

$$v_\alpha^*(s) = v_{\mathcal{E}}^\pi := \max_\pi \min_{Q \in \mathcal{E}} v^{\pi, Q} \tag{20}$$

$$v_\alpha^\pi(s) = v_{\mathcal{E}}^\pi := \min_{Q \in \mathcal{E}} v^{\pi, Q}. \tag{21}$$

$\square$

## C   AUTOEXPECTRL ALGORITHM DESCRIPTION

In the section, we gives implementation details of our algorithm `AutoExpectRL`. First, we choose a neural network that has 4 heads for the critic, one per value of $\alpha$, leading to 4 estimates of the pessimist $Q$-function, $Q_{\phi_d}(s, a), \quad \forall d \in [1, 4]$. Even if some parameters are shared in the body of the network, we denote parameters of the critic as $\phi = \{\phi_1, \phi_2, \phi_3, \phi_4\}$. A similar network is used for actor neural network, with four heads, one per policy $\pi_{\theta_d}, \forall d \in [1, 4]$. with $\theta = \{\theta_1, \theta_2, \theta_3, \theta_4\}$.

Given 4 target $y_d(r, s') = r + \gamma Q_{\phi_{d,\text{targ}}}(s', \pi_{\theta_d}(s'))$ with reward $r$, policy $\pi_{\theta_d}$, we propose to minimize the `AutoExpectRL` critic loss

$$L_{auto}(\phi, \mathcal{D}) = \mathbb{E}_{(s,a,r,s') \sim \mathcal{D}}\left[\sum_{d=1}^4 L_2^{\alpha_d}(Q_{\phi_d}(s, a) - y_d(r, s'))\right]. \tag{22}$$

which as associated `UpdateCritics`$(B, \theta, \phi)$ function which is a gradient ascent using :

$$\Delta_\phi \propto \nabla_\phi \frac{1}{|B|} \sum_{(s,a,r,s') \in B} \sum_{d=1}^4 L_2^{\alpha_d}(Q_{\phi_d}(s, a) - y_d(r, s')). \tag{23}$$

The actor of our algorithm `AutoExpectRL` is updated according to the gradient of the sum of the actor's head losses or `UpdateActor`$(T, \theta, \phi)$:

$$\Delta\theta \propto \nabla_\theta \frac{1}{|B|} \sum_{s \in B} \sum_{k=1}^4 Q_{\phi_k}(s, \pi_{\theta_k}(s)). \tag{24}$$

---

**Algorithm 1** `AutoExpectRL`

---

1: Initialize critic networks $Q_{\phi_d}$ and actor $\pi_\theta \; \forall d \in [1,4]$
  Initialize target networks for all networks, i.e. $\forall d \in [1,4] \; \phi'_d \leftarrow \phi_d, \theta'_d \leftarrow \theta_d$
  Initialize replay buffer and bandit probabilities $\mathcal{B} \leftarrow \emptyset, \mathbf{p}_1^\alpha \leftarrow \mathcal{U}([0,1]^D)$
2: **for** episode in $m = 1, 2, \dots$ **do**
3:     Initialize episode reward $R_m \leftarrow 0$
4:     Sample expectile $\alpha_m \sim \mathbf{p}_m^\alpha$
5:     **for** time step $t = 1, 2, \dots, T$ **do**
6:         Select noisy action $a_t = \pi_{\theta_d}(s_t) + \epsilon, \epsilon \sim \mathcal{N}(0, s^2)$, obtain $r_{t+1}, s_{t+1}$ where $d$ is the index
             in the bandit problem of chosen expectile $\alpha_m$
7:         Add to total reward $R_m \leftarrow R_m + r_{t+1}$
8:         Store transition $\mathcal{B} \leftarrow \mathcal{B} \cup \{(s_t, a_t, r_{t+1}, s_{t+1})\}$
9:         Sample $N$ transitions $B = (s, a, r, s')_{n=1}^N \sim \mathcal{B}$.
10:     `Update Critics`$(B, \theta', \phi')$ according to (23).
11:         **if** $t \mod b$ **then**
12:             `UpdateActor`$(T, \theta, \phi)$ according to (24).
13:             Update $\phi'_d$: $\phi'_d \leftarrow \tau \phi_d + (1-\tau)\phi'_d, d \in \{1,4\}$
14:             Update $\theta'$: $\theta'_d \leftarrow \tau \theta_d + (1-\tau)\theta'_d$
15:     **end for**
16:     Update bandit $\mathbf{p}^\alpha$ weights using : $w_{m+1}(d) = w_m(d) + \eta \frac{R_m - R_{m-1}}{\mathbf{p}_m^\alpha(d)}$
17: **end for**

---

The dimension of our neural network is related to the dimension of the classical network of TD3. First, we choose a common body of share weights for our neural network of hidden dimension $[400, 300]$. Then our network is composed of 4 heads, each with final matrix weights of dimensions $300 \times 1$ where 1 represents the value of one pessimist $Q$-function $Q_k$. The dimension of the actor-network hidden layers is similar to the critic network for share weights, but the non-shared weights between the last hidden layer and the 4 policies have dimension $300 \times |A|$. Finally, the sampling of new trajectories is done using the actor head with the chosen current $\alpha$ proposed by the bandit algorithm using $\pi_{\theta_d}$ with $d$ the index of the chosen expectile.

The algorithm can be summarised as in Algorithm 1. The blue parts are parts that differ from the traditional TD3 algorithm, as they are related to the bandit mechanism or `ExpectRL` losses. Note that the parameter $b$, the delay between the update of the critic and the actor, is usually chosen as 2 in TD3 algorithm. Finally, in the update of the bandit, an extra parameter, the learning rate $\eta$ of the gradient ascent must be chosen. This parameter influences how fast the bandit converges to an arm, and in our case is chosen as $0.2$ like in Moskovitz et al. (2021) which uses bandit to fine-tune parameters in Rl algorithm. for all environments. Finally, in the testing phase of the benchmark, the best arm is chosen to maximize the reward.

## D    Hyperparameters

| Hyperparameter | Value |
|---|---|
| Learning rate actor | $3e-4$ |
| Learning rate critic | $3e-3$ |
| Batch size | 100 |
| Memory size | $3e5$ |
| Gamma | 0.99 |
| Polyak update $\tau$ | 0.995 |
| Number of steps before training | $7e4$ |
| Train frequency and gradient step | 100 |
| Network Hidden Layers (Critic) | [400, 300] like original implementation of TD3 |
| Network Hidden Layers (Actor) | [400, 300] like original implementation of TD3 |

Table 4: Hyperparameters

All experiments were run on an internal cluster containing a mixture of GPU Nvidia Tesla V100 SXM2 32 Go. Each run was performed on a single GPU and lasted between 1 and 8 hours, depending on the task and GPU model. Our baseline implementations for TD3 is Raffin et al. (2021) where we use the same base hyperparameters across all experiments, displayed in Table D.

## E    AutoExpecRL vs other expectiles on Robust benchmark for mean on Table 1

This section illustrates the fact that `ExpecRL` method outperforms on robust benchmark TD3 algorithm. Without any hyperparameter tuning, `AutoExpecRL` achieves a similar performance to `ExpecRL` with the best expectile, finding the best arms in the bandit problem. In Ant and Hopper environments, the best expectile is frequently very low, typically $\alpha = 0.2$ our $0.3$ where this is less the case for HalfCheetah and Humanoid where the best expectile is bigger. Finally, we can remark that smaller expectiles give better performance in terms of min performance while for average metric, higher expectiles are chosen, which is also verified in Table 5 for DR benchmark.

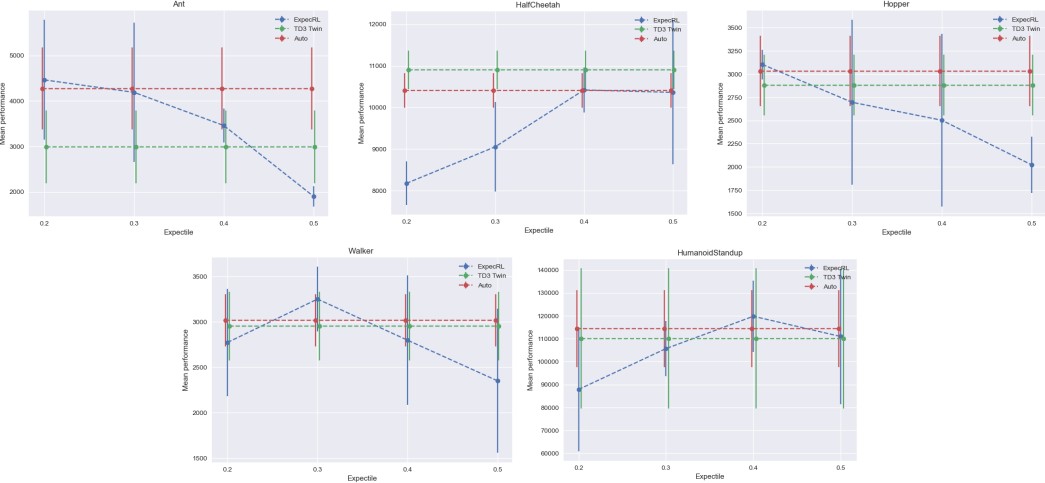

Figure 1: Mean performance as a function of the expectile, non-robust case (corresponding to Table 1).

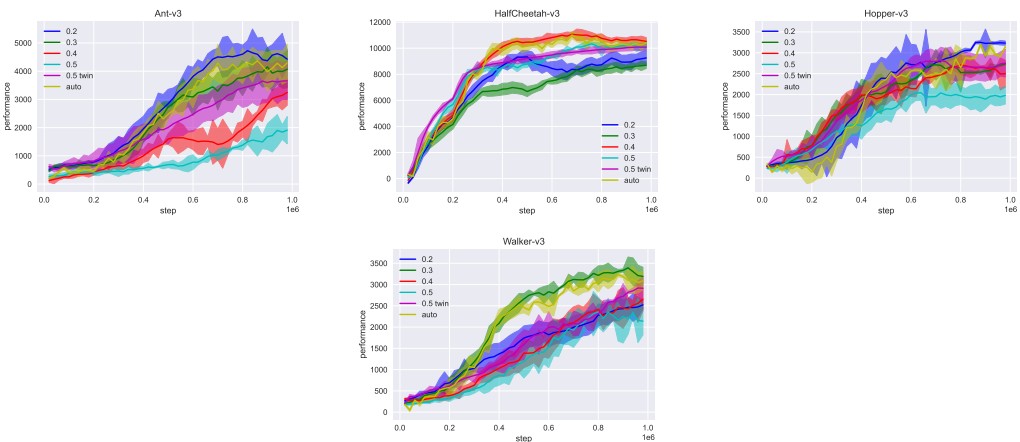

Figure 2: Learning curves non-robust case (corresponding to Table 1).

## F WORST CASE PERFORMANCE FOR AUTOEXPECRL AND EXPECRL (ONLY NOMINAL SAMPLES) OR TABLE 2.

### F.1 FOR 1D UNCERTAINTY GREED BENCHMARK

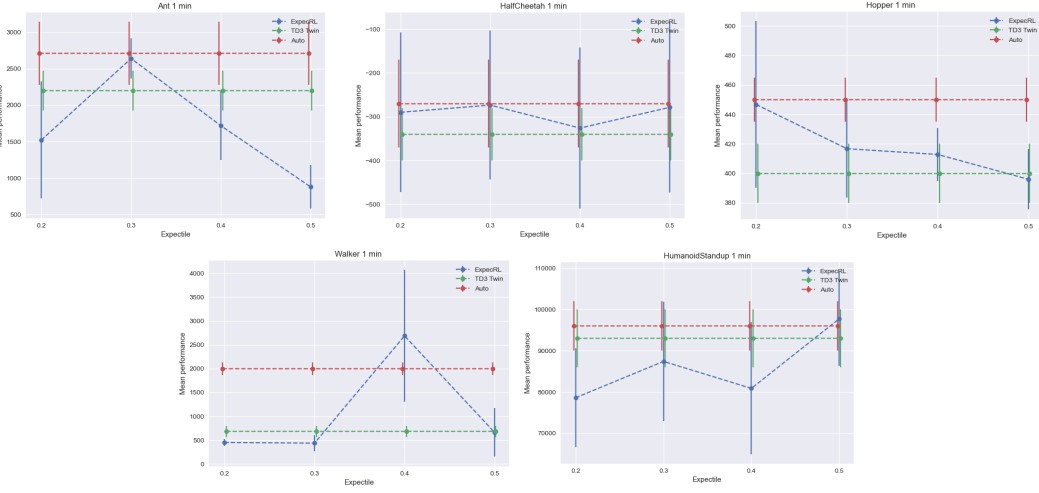

Figure 3: Min performance as a function of the expectile, robust case (corresponding to Table 2).

### F.2 FOR 2D UNCERTAINTY GREED BENCHMARK

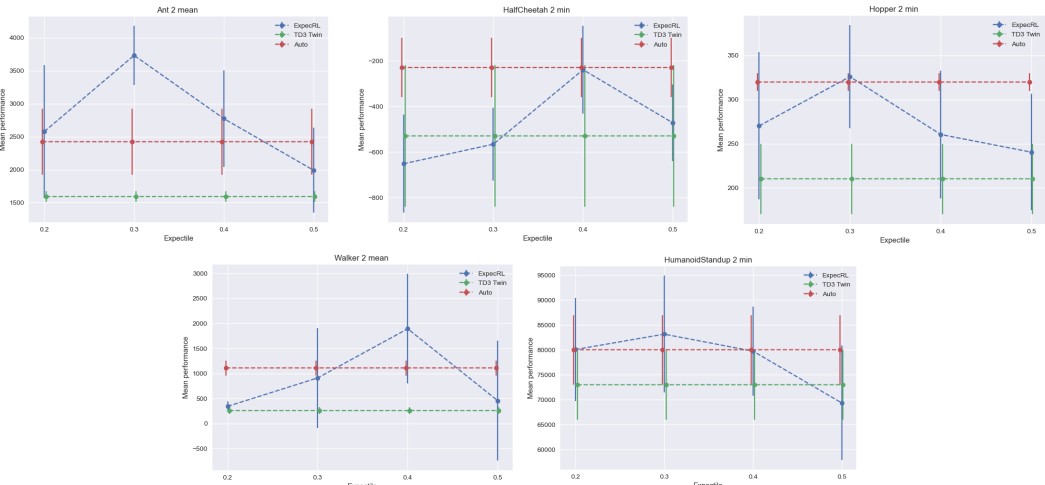

Figure 4: Min performance as a function of the expectile, robust case (corresponding to Table 2).

### F.3 FOR 3D UNCERTAINTY GREED BENCHMARK

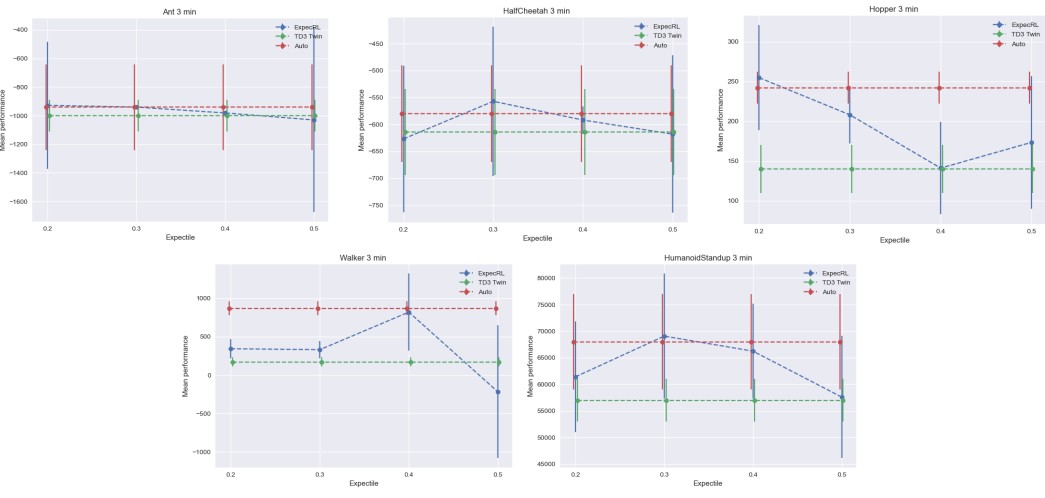

Figure 5: Min performance as a function of the expectile, robust case (corresponding to Table 2).

## G AVERAGE PERFORMANCE FOR AUTOEXPECRL AND EXPECRL(ONLY NOMINAL SAMPLES) OR TABLE 2.

### G.1 FOR 1D UNCERTAINTY GREED BENCHMARK

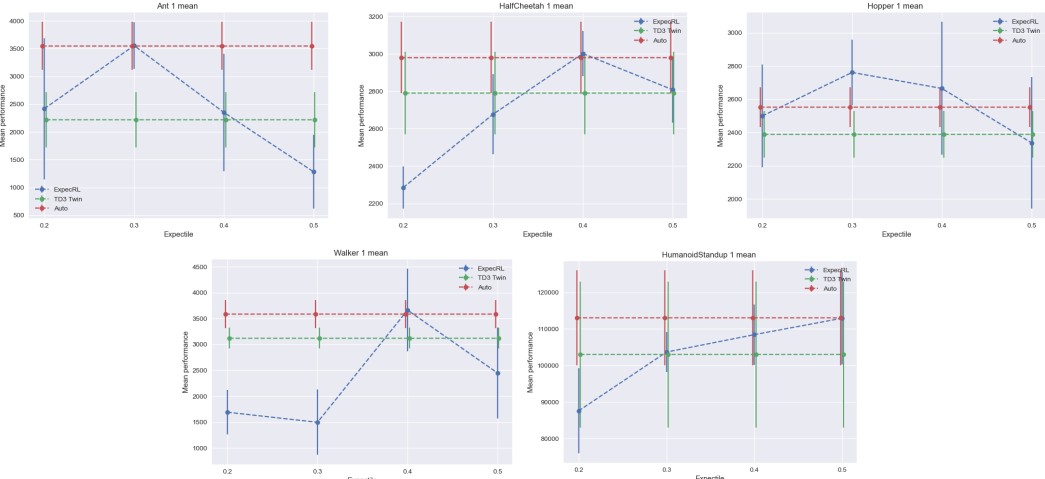

Figure 6: Min performance as a function of the expectile, robust case (corresponding to Table 2).

### G.2 FOR 2D UNCERTAINTY GREED BENCHMARK

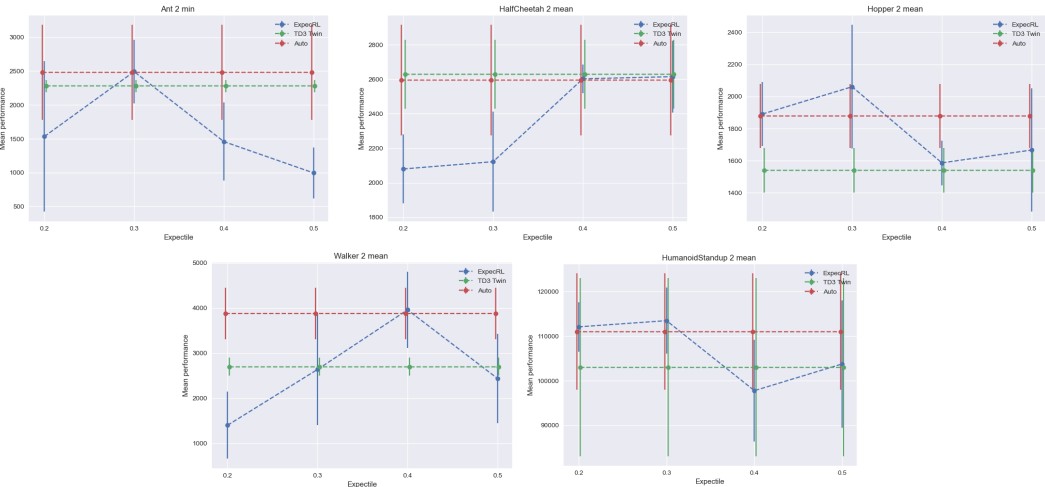

Figure 7: Min performance as a function of the expectile, robust case (corresponding to Table 2).

### G.3 FOR 3D UNCERTAINTY GREED BENCHMARK

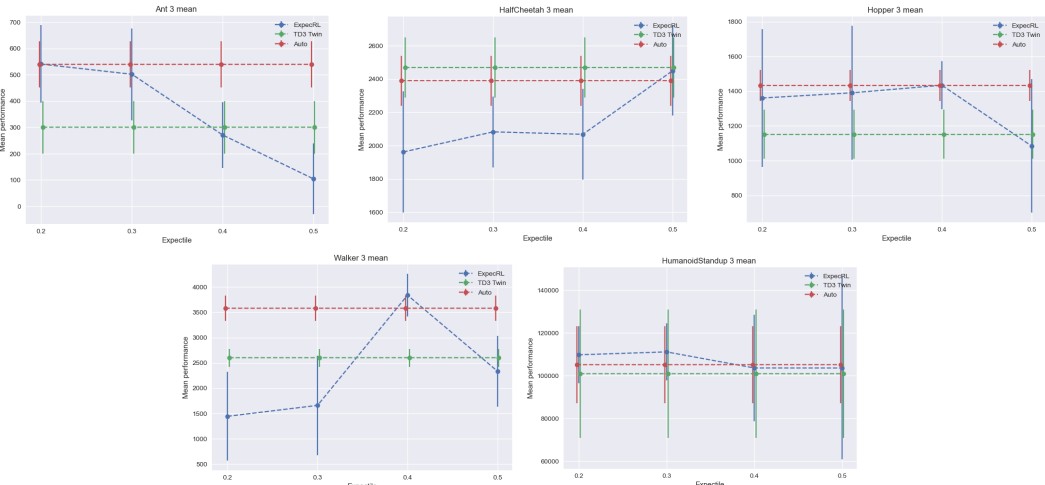

Figure 8: Min performance as a function of the expectile, robust case (corresponding to Table 2).

## H ADDITIONAL DETAILS FOR EXPECTILES ON ROBUST BENCHMARK FOR WORST-CASE AND MEAN ON TABLE 3

| Env | Min | Mean |
|---|---|---|
| $Ant1$ | 3 | 3 |
| $Ant2$ | 2 | 3 |
| $Ant3$ | 2 | 3 |
| $HalfCheetah1$ | 3 | 3 |
| $HalfCheetah2$ | 3 | 3 |
| $HalfCheetah3$ | 3 | 3 |
| $Hopper1$ | 3 | 4 |
| $Hopper2$ | 3 | 4 |
| $Hopper3$ | 3 | 3 |
| $Walker1$ | 3 | 4 |
| $Walker2$ | 4 | 4 |
| $Walker3$ | 3 | 3 |
| $HumanoidStandup1$ | 3 | 3 |
| $HumanoidStandup2$ | 2 | 3 |
| $HumanoidStandup3$ | 2 | 3 |

Table 5: Best Expectile in DR for `ExpectRL`

# I  UNCERTAINTY SETS USED FOR ROBUST BENCHMARK

Table 6: Uncertainty sets used for Robust benchmark

| Environment | Uncertainty Set $\Omega$ | Reference Parameter | Uncertainty Parameter Name |
|---|---|---|---|
| Baseline MuJoCo Environment: Ant | | | |
| Ant 1 | [0.1, 3.0] | 0.33 | torso mass |
| Ant 2 | [0.1, 3.0] $\times$ [0.01, 3.0] | (0.33, 0.04) | torso mass $\times$ front left leg mass |
| Ant 3 | [0.1, 3.0] $\times$ [0.01, 3.0] $\times$ [0.01, 3.0] | (0.33, 0.04, 0.06) | torso mass $\times$ front left leg mass $\times$ front right leg mass |
| Baseline MuJoCo Environment: HalfCheetah | | | |
| HalfCheetah 1 | [0.1, 4.0] | 0.4 | world friction |
| HalfCheetah 2 | [0.1, 4.0] $\times$ [0.1, 7.0] | (0.4, 6.36) | world friction $\times$ torso mass |
| HalfCheetah 3 | [0.1, 4.0] $\times$ [0.1, 7.0] $\times$ [0.1, 3.0] | (0.4, 6.36, 1.53) | world friction $\times$ torso mass $\times$ back thigh mass |
| Baseline MuJoCo Environment: Hopper | | | |
| Hopper 1 | [0.1, 3.0] | 1.00 | world friction |
| Hopper 2 | [0.1, 3.0] $\times$ [0.1, 3.0] | (1.00, 3.53) | world friction $\times$ torso mass |
| Hopper 3 | [0.1, 3.0] $\times$ [0.1, 3.0] $\times$ [0.1, 4.0] | (1.00, 3.53, 3.93) | world friction $\times$ torso mass $\times$ thigh mass |
| Baseline MuJoCo Environment: HumanoidStandup | | | |
| HumanoidStandup 1 | [0.1, 16.0] | 8.32 | torso mass |
| HumanoidStandup 2 | [0.1, 16.0] $\times$ [0.1, 8.0] | (8.32, 1.77) | torso mass $\times$ right foot mass |
| HumanoidStandup 3 | [0.1, 16.0] $\times$ [0.1, 5.0] $\times$ [0.1, 8.0] | (8.32, 1.77, 4.53) | torso mass $\times$ right foot mass $\times$ left thigh mass |
| Baseline MuJoCo Environment: Walker | | | |
| Walker 1 | [0.1, 4.0] | 0.7 | world friction |
| Walker 2 | [0.1, 4.0] $\times$ [0.1, 5.0] | (0.7, 3.53) | world friction $\times$ torso mass |
| Walker 3 | [0.1, 4.0] $\times$ [0.1, 5.0] $\times$ [0.1, 6.0] | (0.7, 3.53, 3.93) | world friction $\times$ torso mass $\times$ thigh mass |

