# OpenReview forum: "Bootstrapping Expectiles in Robust Reinforcement Learning"
_ICLR.cc/2026/Conference — Submitted to ICLR 2026_

### Official Review · Reviewer_qbRC · 2025-10-18

**Soundness:** 3
**Presentation:** 3
**Contribution:** 2
**Rating:** 2
**Confidence:** 4

**Summary:**

This paper proposes ExpectRL, a reinforcement learning algorithm that replaces the standard critic’s L2 loss with an expectile loss to introduce a controllable level of pessimism in value estimation. The authors argue that bootstrapping expectiles instead of expectations leads to a pessimistic Bellman operator, which retains contraction properties and implicitly induces robustness. They further extend this idea to AutoExpectRL, which automatically tunes the expectile parameter $\alpha$ via a bandit-based approach, and to a domain-randomized (DR) setting for robust RL. Empirical studies on MuJoCo benchmarks show that ExpectRL can match or outperform TD3 with fewer critics and competitive performance compared to M2TD3 in robust RL settings, with lower computational and hyperparameter tuning overhead.

**Strengths:**

- The proposed ExpectRL method introduces pessimism in a remarkably simple way — by replacing the L2 loss with an expectile loss. This allows direct integration into existing actor-critic frameworks without major architectural changes.
- The AutoExpectRL component demonstrates a practical way to adapt the pessimism level online without extensive hyperparameter search.
- Mathematically connect the Expectile Bellman Operator to the Robust Bellman Operator

**Weaknesses:**

- The paper claims lower computational costs, but this point are not quantitatively supported.
- The connection between expectiles and robust RL is insightful but somewhat straightforward, given the coherent risk measure foundation.
- Typographical and Presentation Issues:
   - Ensure consistent use of “state-of-the-art” or “state of the art” throughout the manuscript.
   - Figures 1 and 3: x/y labels and legends should be larger for readability.
   - In Table 3, the double-line separator between mean and worst-case results (M2TD3(m) and DR(m)) appears misaligned.
- The current experiments evaluate robustness to environment changes but not to active adversarial perturbations or attacks. To fully support the robustness claim, evaluation under adversarial perturbation settings (e.g., scenarios in RARL) would be valuable.
- Line 466 mentions outperforming state-of-the-art methods such as M2TD3 while it which was published in 2022. However, several more recent robust RL algorithms have been proposed since then (e.g., [1, 2])


#### [1] Juncheng Dong et al., Variational Adversarial Training Towards Policies with Improved Robustness, AISTATS 2024
#### [2] Aryaman Reddi et al., Robust Adversarial Reinforcement Learning via Bounded Rationality Curricula, ICLR 2024

**Questions:**

- Please clarify how algorithms are categorized into the two groups of robust RL methods. Specifically, M2TD3 assumes access to an uncertainty set, whereas RARL-style adversarial training does not require an explicit uncertainty set but learns an adversarial policy. This distinction is important for existing works.
- Could you compare your method with ROSE [1], especially both your paper and this paper aims the balance between optimism and pessimism?
- The claim in Line 164 that “expectile statistic has never been considered before for tackling robust RL” may need revision or clarification. For example, methods such as ROSE [1] and other CVaR-based robust RL approaches already use coherent risk measures to achieve pessimism. Please add discussion clarifying how expectiles differ from or extend these frameworks.
- The distinction between ExpectRL and prior distributional RL approaches using expectiles is not fully clear. Beyond the scope (learning a single expectile vs. the full return distribution) and the application scenarios (robustness), please elaborate the challenge or novelty of the proposed method
- Since both methods use expectile regression in critic learning, please elaborate on the conceptual and algorithmic differences beyond application setting (offline vs. robust online).
- Section 5 shows that different environments prefer different $\alpha$ values. Could the authors provide insights or hypotheses on what environment characteristics and how they influence the $\alpha$ for better results?

If the authors address the above concerns and questions, I would be willing to inclined to increase my overall score.

#### [1] Juncheng Dong et al., Variational Adversarial Training Towards Policies with Improved Robustness, AISTATS 2024

---

### Official Review · Reviewer_9RBp · 2025-10-25

**Soundness:** 1
**Presentation:** 2
**Contribution:** 2
**Rating:** 4
**Confidence:** 4

**Summary:**

The paper introduces ExpectRL, a simple TD-style algorithm that replaces the usual squared Bellman error with an expectile regression loss (τ < 0.5) to inject pessimism, mitigate over-estimation, and improve worst-case robustness. A bandit-based wrapper (AutoExpectRL) is proposed to adapt τ online for standard RL tasks. The method is evaluated on MuJoCo and robust-RL benchmarks, showing competitive or slightly better worst-case returns than twin-critic baselines (TD3, M2TD3) while using only one critic. Theoretical results connect expectiles to coherent risk measures and robust MDPs.

**Strengths:**

- The paper introduces simple and practical change to the critic loss; no twin networks or explicit uncertainty modules.
- Provide theoretical validations that expectiles are related to robust value functions.
- First attempt to adapt the pessimism level $\alpha$ online via a bandit, reducing manual search on stationary tasks.
- Tests include both standard MuJoCo and domain-randomized robust environments; worst-case performance is competitive.

**Weaknesses:**

- Variance of ExpectRL is frequently higher than baselines (Tables 2–3), contradicting the “improved robustness”. Overall, the results are not  impressive enough.
-  Expectiles for pessimism already appeared in LEQ (Park, Kwanyoung et al., Lower Expectile Q-Learning, ICLR 2025), the incremental contribution is not stated. AutoExpectRL actually keeps track of the value function of different $\alpha$s, which is much related with distributional RL methods. Comparison with distributional RL or other risk-sensitive baselines are not provided, either.
-  Pessimism seems to hurt some exploration-sensitive domains (HalfCheetah), and not enough analyses are provided.
-  The paper claimed lower computational cost of ExpectRL, however, no theoretical or empirical computational cost is provided. Especially for AutoExpectRL, it seems it is not necessarily more efficient than TD3.
-  Running ExpectRL with best-of-4 runs is reported without variance correction, inflating apparent gains, which might not be persuasive.

**Questions:**

- Investigation of different $\alpha$ in different tasks might be helpful to analyze the impact of hyper-parameters.
- It would also be beneficial to show how $\alpha$ is adapted in AutoExpectRL for different tasks, especially at different training stages and different tasks.

---

### Official Review · Reviewer_3BCL · 2025-10-30

**Soundness:** 2
**Presentation:** 2
**Contribution:** 2
**Rating:** 4
**Confidence:** 3

**Summary:**

The paper proposes the algorithm ExpectRL that replaces the standard mean-based Bellman update in reinforcement learning with an expectile-based bootstrapping scheme to introduce controllable pessimism. Theoretically, the authors show that this modification is equivalent to solving a distributionally robust MDP, where the expectile parameter $\alpha$ determines the level of robustness. Empirically, ExpectRL achieves competitive or better performance compared to twin-critic TD3 and several robust RL baselines on standard MuJoCo and robust control benchmarks.

**Strengths:**

1. The paper's core idea (replacing the L2 critic loss with an expectile loss) is clean and general, where the loss can be easily integrated into most actor–critic or Q-learning–based methods
2. This paper provides some theoretical interpretation that the expectile Bellman operator is a contraction mapping and is equivalent to solving a distributionally robust MDP.
3. Their algorithm consistently improves robustness and stability across MuJoCo benchmarks. And as author claims the algorithm requires minimum additional hyperparameters tuning.

**Weaknesses:**

1. Lack of analysis on the role of $\alpha$ (expectile parameter). This paper links smaller $\alpha$ to stronger pessimism, but it does not discuss theoretically or empirically how $\alpha$ affects the learned value or final reward, or whether reward varies monotonically with $\alpha$?
2. Similar expectile-based Distributional RL methods are mentioned in related works, but no direct comparison in the experimental part is provided.
3. While the paper proves contraction and equivalence to robust MDPs, there is no more theoretical insights such as sample complexity bounds, convergence rate analysis, or explicit characterization of robustness and $\alpha$ trade-off.

**Questions:**

see weakness. I might increase the score if more baselines or the discussion/experiment on monotone affect of $\alpha$ are provided.

---

### Official Review · Reviewer_TyZ3 · 2025-10-31

**Soundness:** 2
**Presentation:** 2
**Contribution:** 3
**Rating:** 2
**Confidence:** 4

**Summary:**

The paper proposes to use an expectile bootstrapping to achieve controllable pessimism and improved robustness. By using an expectile loss instead of mean squared error, the algorithm naturally reduces Q-value overestimation without needing twin critics. The paper proves that the expectile Bellman operator is a $\gamma$-contraction and corresponds to a robust value function, providing theoretical convergence and robustness guarantees. The paper also proposes AutoExpectRL, which automatically tunes the expectile parameter, to mitigate the hassle of hyperparameter selection. Experiments on MuJoCo benchmarks suggest ExpectRL has an advantage in environments that benefit from pessimistic value estimates.

**Strengths:**

The paper clearly explains the intuition and the methodology. The proposed method is straightforward to integrate with existing methods.

The work is supported by theoretical results. The paper proves contraction properties for an expectile Bellman operator and provides theoretical support for the convergence of the algorithm when $\alpha \leq 0.5$ (Theorem 4.1). Theorem 4.2 shows the optimal value function under expectile setting remains consistent to the optimal robust value function.

Related work is adequately covered, and differences from existing pessimism/robustness approaches are discussed in Section 2, emphasizing the novelty of employing expectile for pessimistic value estimation.

The paper reports experimental settings for reproducible empirical evaluation in Appendix D.

The reported experimental results use multiple runs (10 seeds) with confidence intervals.

**Weaknesses:**

The paper proposes AutoExpectRL as an auto expectile parameter tuning method. However, I am not convinced that the expectile tuning is fully automatic. The expectile value still relies on the manually selected numbers. As the paper introduces on lines 335-336 and lines 344-346, AutoExpectRL runs on a predefined discrete set of expectile values, with both critic and actor employing a multi-head network, and uses different expectile values on each head. The parameter selection problem is transferred from selecting one number to selecting a set of numbers. It may work well in tasks that are not sensitive to expectile setting, but in those that are picky on expectile value, selecting the set of numbers would still require knowledge of the environment; otherwise, there are no guarantees for the method to find the optimal expectile value.

The paper claims that ExpectRL has lower computational costs than other methods on lines 116 and 124-125. Though I understand that using one network should intuitively cost less than double networks, as this point is listed as one contribution, it would make the argument stronger if detailed empirical or theoretical results support it, such as wall-clock time, number of parameters (weights) to learn, and complexity analysis. Besides, the AutoExpectRL introduces a multi-head architecture, which includes more weights to update, while the double critic method is a one-head architecture. In this case, reporting the empirical or theoretical evidence is necessary to indicate that the proposed method (AutoExpectRL) still enjoys a low computation cost.

In online learning scenario, reporting the learning rate is usually more informative than the final performance, as the learning curves show the learning efficiency, stability, and convergence information. But the paper provides the final performance only. As the paper shows the method converges, it would be good to empirically verify it by looking at the late stage in the learning curve.

The advantage of the proposed expectile selection method is not well demonstrated in the empirical test. In Table 2, AutoExpectRL is often worse than ExpectRL. That may suggest an issue in the proposed parameter selection method, and is worth an investigation and further discussion in the paper.

That said, I would be happy to discuss further if I have misunderstood any aspect of the proposed method.

**Questions:**

As the network has shared hidden layers, the loss on each head affects the shared representation and thus the output of other layers. Training in this way means the selected ‘optimal expectile’ may not be the best when using it as the only fixed choice for learning. Could the authors please elaborate on this choice? Does it cause any issues during learning?

---

### Meta-Review · Area_Chair_PbLE · 2025-12-14

**Summary:**

- The paper makes some claims without sufficient support. For instance, it mentions that the approach has a lower computational cost. However, there is no empirical evidence to support this claim.
- The paper focuses on results after training, which limits the evaluation of the algorithm's performance during training.
- Missing analysis of the hyperparameters introduced by the paper.
- Missing clarifications to the questions raised by the reviewers.
- Unclear comparison with pessimistic distributional RL methods

**Reviewer Concerns:**

no rebuttal

**Reviewer Scores:**

no rebuttal

---

### Decision · Program_Chairs · 2026-01-26

Reject